# From Alpha to Omicron: How Different Variants of Concern of the SARS-Coronavirus-2 Impacted the World

**DOI:** 10.3390/biology12091267

**Published:** 2023-09-21

**Authors:** Mickensone Andre, Lee-Seng Lau, Marissa D. Pokharel, Julian Ramelow, Florida Owens, Joseph Souchak, Juliet Akkaoui, Evan Ales, Harry Brown, Rajib Shil, Valeria Nazaire, Marko Manevski, Ngozi P. Paul, Maria Esteban-Lopez, Yasemin Ceyhan, Nazira El-Hage

**Affiliations:** Herbert Wertheim College of Medicine, Biomedical Sciences Program Florida International University, Miami, FL 33199, USA; miandre@fiu.edu (M.A.); llau001@fiu.edu (L.-S.L.); marperry@fiu.edu (M.D.P.); jrame009@fiu.edu (J.R.); fowen008@fiu.edu (F.O.); jsouc004@fiu.edu (J.S.); jakka001@fiu.edu (J.A.); eales004@med.fiu.edu (E.A.); hbrow052@fiu.edu (H.B.); rshil@fiu.edu (R.S.); vnaza002@fiu.edu (V.N.); mmane011@fiu.edu (M.M.); ppaul020@fiu.edu (N.P.P.); meste054@fiu.edu (M.E.-L.); yceyh001@fiu.edu (Y.C.)

**Keywords:** coronavirus, variants of concern, genetic variation, ACE2 receptor

## Abstract

**Simple Summary:**

This review article comprehensively discussed the evolution of the SARS-CoV-2 virus since its initial outbreak in 2019. Specifically, we have provided insights into the genetic mutations and the emergence of different variants of concern and their classifications. We have emphasized the properties of these variants, highlighting their increased transmissibility, virulence, and potential for immune evasion. Lastly, we have provided a glimpse into the future perspective of the global health response and future variant prediction models.

**Abstract:**

SARS-CoV-2, the virus that causes COVID-19, is prone to mutations and the generation of genetic variants. Since its first outbreak in 2019, SARS-CoV-2 has continually evolved, resulting in the emergence of several lineages and variants of concern (VOC) that have gained more efficient transmission, severity, and immune evasion properties. The World Health Organization has given these variants names according to the letters of the Greek Alphabet, starting with the Alpha (B.1.1.7) variant, which emerged in 2020, followed by the Beta (B.1.351), Gamma (P.1), Delta (B.1.617.2), and Omicron (B.1.1.529) variants. This review explores the genetic variation among different VOCs of SARS-CoV-2 and how the emergence of variants made a global impact on the pandemic.

## 1. Introduction

Severe acute respiratory syndrome coronavirus 2 (SARS-CoV-2) has caused more than 200 million infections and about 6.0 million deaths worldwide. SARS-CoV-2 is the seventh coronavirus known to infect humans and the most widely spread coronavirus which caused the COVID-19 pandemic. SARS-CoV and MERS-CoV are two additional coronaviruses that can cause severe symptoms in humans, while others are only known to induce mild symptoms [1]. In terms of genetic make-up, coronaviruses are enveloped, non-segmented, single-strand, positive-sense RNA viruses (+ssRNA) containing a very large RNA (approximately 26–32 kilobases) surrounded by a symmetrical nucleocapsid [2]. The SARS-CoV-2 genome consists of approximately 29,903 nucleotides with at least 10 open reading frames (ORFs) that encode 29 proteins, including 16 nonstructural proteins (NSP), 4 structural proteins (SP), and 9 accessory proteins [3,4]. The first 20 kilobases downstream from the 5′ end encode the ORFs 1a and ab, which encode the nonstructural proteins, with the last 10 kilobases encoding the structural proteins [5,6]. The viral structure includes a nucleocapsid (N) protein surrounded by an envelope comprising a membrane protein (M), spike protein (S), and envelope protein (E) (see Figure 1 below). Perhaps the most important feature of coronaviruses is the heavily glycosylated spike glycoprotein (S, approximately 150 kDa), located on the surface which drives cellular entry [7,8]. SARS-CoV-2 uses an RNA-dependent RNA polymerase (RdRp) comprised of NSP 7, NSP8, and NSP12 to replicate its genome, which has been the target of several antiviral therapies [9]. The majority of coronaviruses utilize a proteinaceous peptidase such as the angiotensin-converting enzyme 2 (ACE2) as a receptor for host cell binding, and in the case of the SARS-CoV-2, binding occurs via the S protein [10,11]. The S protein of the SARS-CoV-2 is cleaved into the S1 (N-terminal) and S2 (C-terminal) subunits by host cell proteases [12]. The S1 subunit acts as the main receptor-binding domain (RBD) and recognizes and binds to the host cell surface receptor. The S1 C-terminal domain (CTD) forms a complex with the human ACE2, whereas the S2 domain is involved in the fusion mechanism between the host cell membrane and the virus [12]. These two subunits (S1 and S2) play a major role in viral infection and pathogenesis and are critical targets for antiviral-neutralizing antibodies. Six RBD amino acids of SARS-CoV-2, including L455, F486, Q493, S494, N501, and Y505 (see Table 1 below), are essential for binding to ACE2 and determining the host range of SARS-CoV-like viruses [1]. The RBD in the spike protein is the most variable part of the coronavirus genome [1].

ACE2 is known to be expressed in proximal and distal airway epithelial cells and studies have shown that ACE2 expression is associated with aging, interferon (IFN) stimulation, electronic cigarette aerosol usage, and is further increased in patients suffering from a chronic obstructive pulmonary disease (COPD) and obesity [30,31,32,33]. ACE2 is additionally expressed in human nasal epithelial cells, lung, spermatogonia, Sertoli, gastric, duodenal, and rectal epithelial cells [34].

Similar to other coronaviruses, SARS-CoV-2 has a high evolutionary rate, of approximately 1.1 × 10^−3^ substitutions per site/year [35]. Since its earlier detection in Wuhan, China, in late 2019, SARS-CoV-2 has undergone many mutations. On 10 January 2020, the first SARS-CoV-2 genome was sequenced and published online. After 13 months, 468,000 sequences of the virus were published [36]. Different SARS-CoV-2 variants have appeared and are driven by a broad spectrum of recombination, point mutations, deletions, and amino acid substitutions, particularly in the S protein RBD of SARS-CoV-2. The World Health Organization (WHO) has classified the SARS-CoV-2 variants into three main categories, such as the variants of concern (VOCs), variants of interest (VOIs), and variants under monitoring (VUMs) [35]. So far, the five reported VOCs include the Alpha (B.1.1.7), Beta (B.1.351), Gamma (P.1), Delta (B.1.617.2), and Omicron (B.1.1.529) variants (Figure 2). VOCs are associated with at least one of the following changes: increased transmissibility; increased virulence; change in clinical disease presentation; or decrease in the effectiveness of public health and social measures, such as available diagnostics, vaccines, or therapeutics [37].

Typically, substitutions and specific deletions have occurred in the S1/S2 domains and RBD areas of genomic sequences in SARS-CoV-2, resulting in the emergence of a new set of SARS-CoV-2 variants. Major (non)synonymous mutations affecting the RBD region in novel SARS-CoV-2 include N501Y, E484K, L452R, and K417N/T [7,38,39]. The D614G substitution in the S protein occurred in all five VOCs and the D614G mutation demonstrates a higher load of infectious virus in the upper respiratory tract and an increased replication and transmissibility in SARS-CoV-2 variants. In the following paragraphs, we have summarized the pathogenicity and critical differences between the current SARS-CoV-2 variants, and how this has impacted the immune response and outcome of infection.

## 2. Alpha (B.1.1.7 Lineage)

### 2.1. Genetic Origin and Genetic Makeup

The Alpha variant (B.1.1.7) has eight mutations in the S protein that affect the conformation of the receptor-binding domain, including an N501Y substitution. Mutations in the N-terminal domain (NTC) include the deletion of two amino acids at positions 69 and 70 (H69/V70) and position 144 (Y144). Researchers showed that the Alpha variant (B. 1.1. 7) S protein can mediate faster syncytium formation, exhibiting enhanced cell–cell fusion activity that is dependent on the ΔH69/V70 deletion [13]. This was an important finding as syncytium formation plays a role in viral replication and is associated with the pathogenesis of severe COVID-19 cases [40,41]. Interestingly, it was found that ΔH69/V70 occurred in multiple independent lineages, after the acquisition of N439K and Y453F, both of which aid antibody evasion, however ΔH69/V70 itself was found to increase infectivity and enhance the incorporation of cleaved S protein into virions, leading to faster viral kinetics compared to the wild-type virus [13]. The N501Y mutation was found to aid viral binding to the ACE2 receptor and to significantly contribute to viral proliferation and infectivity [14]. This deletion causes S-gene target failure in the RT-PCR-based diagnostic assays [15]. 

In addition, the Alpha variant (B.1.1.7) upregulates key viral RNA and proteins responsible for suppressing the innate immune response in airway epithelial cells much more effectively than the earlier observed variants. Orf9b, Orf6, and nucleocapsid protein RNA and protein levels are enhanced in this strain, causing a delay of early host innate response to this variant. These proteins are considered innate immune antagonists, as they actively suppress the innate immune system. 

### 2.2. Epidemiology and Morbidity

In September 2020, the first VOC was detected in Kent, United Kingdom, and was subsequently termed the Alpha variant (Figure 3), also known as B.1.1.7. The Alpha variant (B.1.1.7) spread quickly and by February 2021, it was the predominant variant in England, accounting for over 95% of cases. Importantly, the rise in this variant also was associated with a rise in symptomatic cases and hospitalizations, resulting in an increased strain on the healthcare system [42]. In England, researchers analyzed 2,245,263 positive SARS-CoV-2 community tests and concluded that the Alpha variant (B.1.1.7) had a hazard of death 55% (95% CI 39–72%) higher than the original SARS-CoV2 virus [43], showing that the Alpha variant had both high transmittance and virulence. About 189 countries have reported the Alpha variant [13]. After September 2021, the Alpha variant has not been observed [13].

### 2.3. Response to Treatment/Vaccination 

The Alpha variant resulted in great concern, as the extensive mutations to the spike gene in B.1.1.7 could lead to antigenic changes that could be harmful to monoclonal antibody therapies and vaccine protection. Multiple studies state that B.1.1.7 is resistant to neutralization by monoclonal antibodies directed against NTD supersite [15]. However, despite the strain the Alpha variant placed on the healthcare system and the concern it raised about causing vaccines to become less effective, other variants, such as the Delta variant, soon emerged. Immediately, scientists observed that the Delta variant would become the predominant variant in patients presenting with mild to moderate infections and in fully vaccinated individuals presenting with symptoms [42]. With the rise of other variants, such as Delta and Omicron, the Alpha variant became extinct. 

## 3. Beta (B.1.351 Lineage) 

### 3.1. Genetic Origin and Genetic Makeup

Every variant shares a D614G mutation in the S protein. In the case of the B.1.351 variant, mutations in the S protein have been identified in the RBD (N501Y, E484K, and K417N), the NTD (L18F, D80A, and D215G), and in loop 2 (A701V) [16]. The E484K mutation conferred resistance of the Beta variant to monoclonal antibodies targeted against the receptor binding motif (see Table 1) [16,17,18,19]. E484K and K417N mutations have been shown to affect neutralizing antibody binding ability. In the Beta variant, the K417N mutation in conjunction with the D614G mutation created a more open spike trimer, conferring ACE2 receptor binding characteristics [44,45]. Mutations that provided a more open conformation in the Beta variant S protein prevented the salt bridge/hydrogen bond network from forming, hence preventing the formation of the more closed conformation seen in the Wuhan strain of virus S protein [45]. Despite these conformational differences between the Beta variant and the wild-type (WT) Wuhan strain in its binding characteristics to ACE2, the Beta variant does have lower fitness compared to the Alpha variant in terms of replication kinetics in human airway epithelial cells in culture. The replicative capacity of the Beta variant was significantly lower than that of the Alpha or the progenitor variant when tested in hamster models which support the high replicative capacity of these viruses. Therefore, fundamental distinctions are importantly drawn where Beta and Alpha variants may have enhanced binding capacity to the ACE2 receptor, but Alpha may be more capable of replicating than Beta variants once cells have been infected [46].

It has been reported that the B.1.351 variant, particularly the variant with N501Y mutation, is associated with an increased transmissibility [47], which could explain its rapid spread and the dominant circulating variant in South Africa by December 2020. Furthermore, this variant has also been reported worldwide. However, based on reports from the Centers for Disease Control (CDC), despite increased infectivity and neutralizing antibody evasion, there has been no report of impact on disease severity by the B.1.351 variant. In a K18-hACE2 transgenic mouse study, it was shown that Alpha and Beta variants were 100-fold more lethal than the original SARS-CoV2 [48]. Several studies have shown that convalescent-phase sera and monoclonal antibodies are less effective against the Beta and Omicron variants (discussed below) as compared to the Alpha variant. 

### 3.2. Epidemiology and Morbidity

The B.1.351 variant was originally identified in South Africa as it became the major variant infecting the population (Figure 2). This variant has been reported to have higher transmissibility, characterized by many mutations predominantly located in the spike protein. During the Beta wave of COVID-19 infections in South Africa, it was reported that 12.6% of those who were infected became hospitalized [49]. Out of those, 63.4% had severe disease, with a 28.8% fatality rate. Comparisons between the Alpha and the Beta variants with parameters, including 60-day mortality, 28-day mortality, need for mechanical ventilation, or mechanical ventilation duration, showed no significant differences. 

### 3.3. Response to Treatment/Vaccination 

Despite the severity of the disease when contracting the Alpha and Beta variants of SARS-CoV2, a single dose of vaccines such as the mRNA-1273 vaccine was shown to be highly effective against B.1.1.7 and B.1.351 infections, COVID-19 hospitalization, and death, regardless of whether patients develop symptoms or not [50]. Two doses of the BNT162b2 vaccine also produced T-cell response by epitopes that are not different between variants which include B.1.351 [51]. It is important to highlight that when measuring the efficacy of a vaccine against a specific variant, BNT162b2 was 75% percent effective against the B.1.351 variant [52]. Despite its lower vaccine effectiveness against the B.1.351 variant compared to other variants in the study such as B.1.1.7, the protection of the vaccine against the most severe forms of the disease such as hospitalization and death is still extremely robust, as reported with greater than 90% protection. Novavax also produces vaccines that showed 86% protection against the UK variant and 60% protection against the South African variant [53].

## 4. Gamma (P.1 Lineage)

### 4.1. Genetic Origin and Genetic Makeup

The Gamma variant has a total of 17 mutations when compared to the earlier VOCs, including three that affected the S protein (K417T, E484K, and N501Y). According to the epidemiological data, this further suggested that these residues are associated with an increased binding to the human ACE2 receptor. As supported by the two-category dynamical model, which integrated genomic and mortality data, it showed a 1.7- to 2.4-fold increase in transmissibility and higher viral load than previous COVID-19 infections (non-P.1) [20]. 

### 4.2. Epidemiology and Morbidity

The Gamma variant, also known as P.1, was classified as the third VOC [54]. Initially discovered between November and December 2020 in Manaus, Brazil (Figure 3), this variant spread rapidly to more than 36 countries [20]. During that period, a second wave of COVID-19 evolved and was associated with a significant rise in new cases and deaths. This was stimulated by the emergence and circulation of several variants under monitoring (VUMs), such as the P.2 (i.e., Zeta) and Gamma (i.e., P.1); the latter became widespread by January 2021. 

Despite the national vaccination roll-out beginning on 17 January 2021, the COVID-19 death toll in the country steadily rose in March 2021, reaching a peak in April 2021 of 4.249 deaths/day. Closely followed by a decrease in the number of daily cases and deaths. Zeta (P.2) minimally dominated the initial epidemic wave persisting up to March 2021, but it was rapidly replaced by Gamma (P.1) [55]. This was characterized by an upsurge in total case numbers with the peak observed between February and June 2021. Very rapidly, the Gamma variant became responsible for 96% of cases in Brazil until it was replaced with the highly transmissible Delta variant [55].

### 4.3. Response to Treatment/Vaccination 

Previous infection provided between 54 to 79% protection against P.1. This has been further confirmed by pseudotyped neutralization assays of anti-SARS-CoV-2 RBD monoclonal antibodies, as it showed a reduction in neutralization potency of plasma from vaccinated or naturally infected individuals [19]. Overall, the threat of increased re-infection or decreased vaccine protection posed by P.1 may not be as severe as B.1.351, and current vaccines offer only a medium amount of protection especially with new VOCs constantly evolving [56].

## 5. Delta (B.1.617.2 Lineage) 

### 5.1. Genetic Origin and Genetic Makeup

The fourth and one of the most prominent SARS-CoV-2 VOC emerged in October 2020 in Maharashtra, India, and was termed B.1.617.2 (Delta variant). This variant was shown to have a mutation in the P681R furin cleavage site (FCS). This mutation led to a novel increase in pathogenicity and made the Delta variant more fusogenic, which accounted for an increased infection severity as well as the presence of unusual symptoms [57]. In addition, mutations in the S protein region D950N, T19R, P681R, G142D, D164G, T478K, R158G, Δ156-157, Δ213-214, L452R (see Table 1), were also detected [21]. Compared to the original SARS-CoV-2 variant, the Delta variant has a total of 656 unique mutations, while the B.1.617.2.1 or AY.1 variant (delta plus variant) has 269 mutations. Despite a higher total number of mutations, the Delta variant has 29 highly prevalent mutations in comparison to the 40 highly prevalent mutations in the Delta plus variant. Highly prevalent mutations have a prevalence of >20%. When the Delta variant was compared genetically with the Delta plus variant, Delta plus had exclusive S protein mutations in positions V70F, W258L, and K417N, and mutation in V70F had a 52% prevalence in the Delta plus variant [58]. Moreover, the Delta plus variant has a mutation in A1146T, found in ORF1a. The most prevalent mutation in the Delta variant is D164G, which is known to increase the fitness of the SARS-CoV-2 virus by increasing virion stability in vivo. The combination of deletions of D614G and E484Q in the Delta variant granted the virus a significant increase in infectivity and transmissibility and empowered the virus as the response rate to monoclonal antibodies and targeted vaccines reduced substantially [59]. This allows the virus to better replicate in the epithelial cells of the lungs [21,60]. Intrinsic disorder prediction of both the Delta and Omicron variants identified disordered areas of the virus, specifically within the S protein. The screen identified disorder in amino acid residues 469–471 of the RBD of the Delta variant S protein but not in the Omicron RBD. This level of disorder identified in the Delta variants RBD could influence the stability of the spike protein and its binding affinity to the ACE2. Moreover, the global replacement of the G614 variant with the D614 variant in SARS-CoV-2-infected individuals showed a shift to a more dominant form of the pseudovirus. According to Korber et al., spike G614 in human patients was associated with greater infectivity and higher levels of viral nucleic acid in the upper respiratory tract, despite lower RT-PCR cycle thresholds, suggesting higher viral loads, but not with increased disease severity [61]. This variant has further increased resistance against neutralizing antibodies, and specifically, the RBD E484K mutation was identified to significantly contribute to this ability in both Brazilian as well as South African isolates [62]. The emergence in Delta is attributed in part to the increased affinity and ACE2 binding. However, it is also attributed to increases in other aspects of viral fitness. Like the Kappa variant, Delta was found to have increased antibody neutralization escape, both by monoclonal antibodies as well as vaccine-induced polyclonal antibodies. However, the increased RBD-ACE2 binding also plays a significant role in viral escape [63]. 

### 5.2. Epidemiology and Morbidity

The Delta variant has been subsequently reported in over 119 different countries and served as a prominent driver of the COVID-19 pandemic global spreading, despite the already introduced vaccination efforts that were taking place on a global scale [64]. As the leading global VOC from late 2020 to early 2021, the Delta variant accounted for an estimated 99% of total SARS-CoV-2 cases. The Delta variant was preceded by the B.1.617.1 (Kappa) variant. Delta gained dominance over the Kappa variant due to its increased infectivity, subsequently leading to it becoming the major SARS-CoV-2 lineage globally [63]. Epidemiological studies have shown that the Delta variant is estimated to be more than 40–60% more transmissible than the original SARS-CoV-2 variant [65]. These data align with the predicted mean basic reproductive number (*R*_0_) of the Delta variant, which is 5.08 when compared to the Alpha variant with an *R*_0_ value ranging from 4 to 5 or the ancestral SARS-CoV-2 strain of 2.79 [22,66]. Reports have shown that vaccination, including mRNA-based vaccination, can reduce the chances of hospitalization or death by the SARS-CoV-2 Delta variant by 95% [67]. However, it was noted that increased time after primary immunization dose correlates with decreasing protection against infection from the Delta variant, as well as other variants [68]. The decrease in vaccine efficacy against the prevention of mild SARS-CoV-2 symptoms correlated with a decrease in host antibody levels [69].

### 5.3. Response to Treatment/Vaccination 

Vaccines approved for the prevention of SARS-CoV-2 infection include two mRNA vaccines produced by Moderna, mRNA-1273, and Pfizer-BioNTech, BNT162b2, and an adenoviral vector vaccine produced by Johnson & Johnson’s, Janssen. Moreover, as of July 2022, the mRNA Pfizer-BioNTech vaccine was the only SARS-CoV-2 vaccine approved for administration in children ages five to seventeen [23]. Data from a negative case-control study with human subjects suggested the ChAdOx1 or BNT162b2 vaccine against the Delta variant has considerably lower effectiveness after an initial dose of 30.7% at preventing symptomatic disease when compared to 48.7% effectiveness against the Alpha variant. Data post two immunizations of collated vaccine data from the ChAdOx1 and BNT162b2 showed much less variable with 79.6% against the Delta variant and 87.5% against the Alpha variant [70].

Vaccines developed against SARS-CoV-2 are highly effective at preventing the development of severe disease and hospitalization up to six months post-immunization. However, passable decreases in vaccine effectiveness and efficacy against infection have been seen with the Beta, Delta, and Omicron variants 20 weeks post-secondary vaccination [70]. Additionally, an increase in time after a primary immunization dose negatively correlates with protection from infection especially against infection from the Delta variant due to decreased antibody titers [70]. These decreases in vaccine efficacy at the prevention of mild disease are most likely caused by a decrease in the host immune response induced by the vaccine and not due to an increase in the Delta variants prevalence, as detailed through a meta-regression analysis [69]. 

## 6. Omicron (B.1.1.529 Lineage) 

### 6.1. Genetic Origin and Genetic Makeup

Over 60 mutations, consisting of substitutions, insertions, and deletions have been identified, of which over half were accumulated in the S protein [71]. The number of mutations in the Omicron S protein outnumbered those observed in the above-mentioned VOCs by 3–4-fold [72]. Several mutations, including Δ69–70 deletion, T95I, G142D/Δ143–145 deletion, K417N, T478K, N501Y, N655Y, N679K, and P681H overlapped with those of the other four VOCs described above [24]. The S protein RBD of the Omicron variant possesses 15 mutations, whereas that of the Delta variant possesses the L452R and T478K mutations [72]. These mutations are the most concerning because they have an impact on host immunity, viral entry, rate of infectivity, and transmissibility of the virus [25]. 

#### Omicron Sub-Lineages (BA.1, BA.1.1, BA.2, BA.4, BA.5, BQ.1, BQ.1.1, and XBB.1.5) 

There are more than 50 mutations dispersed throughout the genome that have been identified for BA.1 sub-lineage and the *spike* gene carries 30 substitutions plus the deletion of 6 and insertion of 3 residues [73]. Many of these variants are in the RBD and NTD regions [74]. Of the 19 unique mutations of BA.1 not detected in other variants, 13 are detected in the S protein [73]. The RBD contains mutations in S371L, G446S, and G496S. All these mutations have the potential to differentially affect antibody binding and could modulate neutralization, particularly BA.1 G446S, G496S which lie at the edge of the ACE2 binding footprint. Interestingly, the sub-lineage specific mutations segregate, with BA.1 and BA.1.1 changes lying on one side of the ACE2 footprint and BA.2 changes on the other side, possibly reflecting different selective pressure on the BA.1 and BA.2 sub-lineages [75].

As a sub-lineage of BA.1, BA1.1 harbors a very similar genetic makeup to BA.1. Compared to the other Omicron variants, BA.1 and BA.1.1 showed less transmissibility, global dominance, re-infections, contagion, hospitalization, but more deletions in the S protein [76]. The differentiating factor between these two variants is a unique R346K substitution in the S protein [77]. A total of 40 mutations have been identified in BA.1.1, and it was shown to have higher binding strength to ACE2 than BA.1. The R346K mutation in the RBD of BA.1.1 enhanced the interaction with hACE2 through long-range alterations [78]. Unique mutations relative to the remaining Omicron variants that are shared between BA.1 and BA.1.1 included the A67V, 69–70, T95I, 143–145, N211I, 212, S371L, G466S, G496S, T547K, N856K, and L981F [79]. Data collected by others suggested that BA.1.1 is less severe but more infective than the previous variants [80]. Omicron sub-lineages BA.1 and BA.1.1 displayed significant immune escape, making them potentially as dangerous as previous variants [81].

There was a rapid surge in the proportion of BA.2 globally at the start of 2022. A study showed that the BA.2 sub-lineage was linked with a higher secondary attack rate and an increased susceptibility to infection in both unvaccinated and vaccinated individuals compared to the BA.1 [81]. Although BA.2 is considered an Omicron variant, its genomic sequence is very different when compared to BA.1, suggesting differences in the virological characteristics between the two [77]. According to sequencing data, the BA.2 lineages have 51 mutations dispersed throughout its genome, of which 32 are shared with BA.1. Moreover, the BA.2 sub-lineage has 19 signature mutations and compared with BA.1 and BA.1.1, the S protein mutations in BA.2 is very different [79]. Among the 32 common mutations between the sub-lineages, 21 were detected in the S protein, while the remaining 11 were detected in four coding regions ORF1ab, E, M, and N [73]. BA.2 has an additional 3 deletions and 7 substitutions compared to BA.1, specifically, in the RBD, BA.2 carries four unique mutations: S371F; T376A; D405N; and R408S. Accordingly, the mutations in D405N and R408S have the potential to differentially affect antibody binding and could modulate neutralization due to the location at the edge of the ACE2 binding footprint [75]. BA.2 shares the signature Delta variant mutation (G142D) believed to be linked with a high infectivity [82]. BA.2 may have developed novel mutations to maintain receptor binding like Wuhan-Hu-1 (WT), and to better evade antibody binding, as compared to BA.1 [82]. 

Studies have shown that vaccine-induced humoral immunity does not function against BA.2 with the same efficacy as with BA.1, as a result, higher viral RNA loads in the lungs and lung periphery of BA.2-infected animals were detected as compared to animals infected with BA.1. BA.2 showed greater infectivity, about 1.4-fold higher as compared to the BA.1 lineages and exhibited a higher fusogenicity than its predecessors. In cell culture experiments, BA.2 showed a greater replication rate in human nasal epithelial cells and more fusogenicity than the BA.1 lineage. The BA.2 sub-lineage remained the dominant Omicron variant, albeit declining from 78% to 75% by the end of 2021. Moreover, Omicron sub-lineages BA.1, BA.1.1, and BA.2 were comparably neutralized by Omicron patient sera, reinforcing the importance of booster vaccine doses [81].

In the Spring of 2022, BA.4 and BA.5 were the newest members of Omicron’s growing family of coronavirus subvariants that have been detected in dozens of countries worldwide. The two variants are more like BA.2, although both BA.4 and BA.5 carry their unique mutations, including changes called L452R and F486V in the viral S protein. The BA.4 and BA.5 subvariants were able to spike globally because they could spread faster than other circulating variants. Both BA.4 and BA.5 could evade host immune responses, and they were not significantly impacted by previous vaccines. According to laboratory studies by others, antibodies triggered by vaccination were less effective at blocking BA.4 and BA.5 than they were at blocking earlier Omicron strains, including BA.1 and BA.2 [83,84]. This means that people who received vaccination and were previously infected with Omicron do not have the immunity to defeat the BA.4 and BA.5 variants. Research teams have attributed that to the variants spike mutation in L452R and F486V [85]. According to the research by Sato et al., they found that BA.4 and BA.5 were deadlier in hamsters when compared to the BA.2 sub-lineage, and they were better able to infect cultured lung cells [86]. BA.5 became the predominant virus strain in the U.S., only to be replaced in November 2022 by two new subvariants known as BQ.1 and BQ.1.1. At the beginning of 2023, a new subvariant called XBB.1.5 nicknamed the Kraken, derived from the BA.2 Omicron subvariant was on the rise. The Omicron XBB.1.5 has a spike RBD change, F486P allowing enhanced binding of SARS-CoV-2 to the ACE2 receptor, which may be an important factor in its transmission and infection rates [87]. 

### 6.2. Epidemiology and Morbidity

The original Omicron strain was first identified in Botswana and South Africa in late November 2021, and cases quickly began to surface and multiply in other countries. Omicron was soon classified as the fifth VOC [88]. Phylogenetic analysis of the Omicron variant showed that it originated from the B.1.1.519 (20B) lineage. Based on the S protein sequence, the Omicron variant was divided into two subclades: Subclade 1 had a low-sequence frequency and is found mostly in South Africa. Subclade 2 on the other hand is found globally at high frequency [71].

During the Omicron-predominant period, the weekly COVID-19-associated hospitalization surveillance network (COVID-NET) reported rates per 100,000 adults aged ≥18 years revealed a peak at 38.4 for Omicron, compared with 15.5 during Delta predominance [89]. By December 2021, Omicron was causing daily case numbers in the U.S. to reach over a million, and it began to spawn subvariants. 

Comparison data between SARS-CoV-2 WT (from Wuhan) against the Alpha, Beta, Delta, and Omicron variants in a transgenic mouse model, supported an attenuating trend for the emerging variants, with the Omicron variant being the mildest. This same study showed that the Omicron variants have impaired virus replication in human and animal cells when compared to the WT viral strain, suggesting that Omicron infections in the human population may potentially result in milder lung symptoms compared to those after infection with previous variants. However, it should be noted that animal and cell line studies may not be an accurate representation of the human lungs [90]. 

In addition, the Omicron variant had a shorter median length of hospital stay, fewer intensive care unit admissions, invasive mechanical ventilation, and death [80]. Although the Omicron variant had a replication rate in the lungs that was 10 times lower when compared to the Delta variant, it replicated around 70 times higher in the human bronchus when compared to the Delta variant and the original SARS-CoV-2 virus [91]. Due to the high mutation rate, the efficacy of neutralizing antibodies and vaccination has been reduced, which led to an increased rate of re-infection with the Omicron variant in South Africa [26]. On the other hand, in a study 51 individuals of which 88% were vaccinated and 92% used masks were contacted by a pre-symptomatic physician. Among the 51 contacts, only one person was infected with the Omicron variant, suggesting that the gravity of vaccination and preventive measurements is effective [92].

In both the US and South Africa, the case fatality ratio with the Omicron variant was two-fold less than with the Delta variant [93]. A mathematical model developed to estimate the transmissibility and infection fatality ratio of the Omicron variant in South Africa showed that despite the higher transmissibility rate, the infection fatality was reduced by 78.7% when compared to the previous variants [94]. A patient cohort study from South Brazil showed consistent findings with that from South Africa and the US. In one week, about 1.3% of the state’s population was infected by the Omicron variant, indicating a high transmission rate. However, the lethality was substantially reduced due to the high vaccination rate (70%) in the population [95].

In pediatric patients, the admissions to hospital due to the Omicron variant increased, although the case fatality ratio was reported to be similar to or less than the Delta variant in the adult population [96]. In South Africa, the case fatality ratio for children under the age of 5 was reported to be 0.5% with the Omicron versus 0.6% with the Delta variant [49]. Likewise, in the United Kingdom, the portion of ventilated children in the same age group was reported at 2.9% with the Omicron variant and about 1% with the other variants. Due to the multiple mutations in the S proteins the Omicron variants have increased affinity binding to host ACE2 receptors and have become more transmissible. The mutations also allowed these variants to escape the immune system antibodies. 

Data available up to March 2022, showed that the Omicron variant was having an increased incidence worldwide over the Delta variant, and infections with the Omicron variant caused the lowest reduction in body weight and mortality rates [90]. Other studies showed that the Omicron sub-lineages, especially the BA.1 and BA.1.1 strains, exhibited substantial immune escape that was largely overcome with the mRNA vaccine boosters [81]. However, the variants of Omicron were less severe than the Delta variants, and no differences in differential risk of severe outcomes were detected between BA.1, BA.1.1, and BA.2 [97]. According to the CDC, the BA.2 sub-lineage became the most frequent variant [98]. Fortunately, as of June 2022, global Omicron infection has declined from their all-time highs, with the BA.1 sub-lineage specifically having declined in prevalence from 7% to 4%. 

In December of 2022, the XBB.1.5 subvariant caused less than 10% of COVID-19 cases in the U.S. which rapidly increased to about 40% of COVID-19 cases by January 2023 [87]. Currently, the XBB.1.5 subvariant accounts for about 43% of cases, making it the most dominant strain in the country, with the BQ.1.1 subvariant reporting as the second most dominant strain, with about 29% of COVID-19 cases. The XBB.1.5 subvariant is the most dominant strain in the northeast part of the country, making up more than 80% of cases of COVID-19 in New York and New Jersey. The rapid spread of this Omicron subvariant, XBB.1.5, and the predicted increase in cases may be explained by its mutation lineage. XBB and XBB.1 have shown the highest levels of immune escape among all the Omicron subvariants currently identified and have shown significant reductions in the neutralizing capacity of serum from vaccinated individuals. XBB.1.5 might have these properties. According to a report by the WHO, between October 2022 and January 2023, 5288 sequences of the Omicron XBB.1.5 variant were identified in 38 countries, and most of these sequences were from the US (82.2%), followed by the UK (8.1%), and Denmark (2.2%) [99]. XBB.1.5 was further detected in other countries but not as a dominant strain. XBB.1.16—dubbed “Arcturus” on social media—is another descendant of Omicron detected in early January and most cases have been seen in India. Since XBB.1.5 and XBB.1.16 carry a similar genetic make-up, experts believe that both variants will respond well against the COVID-19 vaccines available on the market. Recently, the WHO has designated XBB.1.5 and XBB.1.16. as variants of interest (VOI) [37]. It is worth noting that two other XBB strains, XBB.1.9.1 and XBB.1.9.2, are rising in the U.S. Together, the two strains account for about 11% of new cases, according to the CDC. Meanwhile, experts are still learning about several newer Omicron strains circulating in the U.S., including BF.7, BN.1, and BF.11. The subvariant BF.7 (BA.2.75.2) has the highest number of reported cases, accounting for 76.2% of all cases worldwide [100]. 

### 6.3. Response to Treatment/Vaccination 

COVID-19 vaccines are a core prevention strategy for severe illness, hospitalization, and death against COVID-19. However, despite aggressive public health campaigns and widespread vaccine availability in the United States, hesitancy persists. The causes for vaccine hesitancy are multifactorial and poorly understood. An overabundance of information, both online and offline, about conspiracy theories on the origins of the virus, as well as suspicions around the motives behind government COVID-19 control measures, has implanted a mistrust in the vaccination intent. In the U.S.A., unvaccinated patients had a higher death rate compared to the vaccinated [101]. The vaccine is deemed one of the best treatments to prevent COVID-19. However, as new variants emerge, it becomes questionable how effective these vaccines are at neutralizing the variants. Studies have shown that the Omicron variants have the potential to escape from neutralizing antibodies. A combination of a populace with largely waning vaccine-induced immunity and increased viral virulence contributed to the striking global rise in COVID-19 infections coincident with the increasing prevalence of B.1.1.529 starting around December 2021 [27]. Vaccine-elicited humoral immunity was specifically challenged by an intrinsic viral antigenic escape. This phenomenon is consonant (and was predicted) with the elapsed time since the initiation of the global vaccination effort. Given the survival challenge applied by vaccination to the voraciously replicating virus, mutations naturally drift the structure of the evolving virus away from the most prevalent antigenic structures [26]. Two doses of both the ChAdOx1 nCoV-19 or BNT162b2 vaccines alone have been reported as ineffective in protecting against symptomatic disease against the B.1.529 variant. Further, although a “booster” dose did indeed decrease susceptibility to infection, this benefit was transient [68]. Those infected with the B.1.1.529 variant showed less lethality, although the modalities available for treating those who were ultimately severely affected by Omicron were quite restricted. Despite B.1.1.529 arising well over a year after the onset of the global pandemic, consistently beneficial treatment of hospitalized victims was still largely limited to glucocorticoids. Indeed, therapeutic monoclonal antibodies utilized in clinical practice were specifically illustrated to be mostly ineffective in therapeutic monoclonal antibodies [28]. Hospital studies conducted among persons not fully vaccinated reported 569,142 (92%) COVID-19 cases, of which 34,972 (92%) resulted in hospitalizations, and 6132 (91%) resulted in COVID-19–associated deaths, while among 46,312 (8%) cases of vaccinated individuals, 2976 (8%) reported hospitalizations, and 616 (9%) reported deaths, suggesting that treatment in patients with any COVID strain benefit from proper vaccination [29]. Studies have shown that the Omicron variants can escape from neutralizing antibodies, questioning the vaccine doses needed to achieve maximum efficacy for these new Omicron variants. In a study by Ai et al., the results indicated that two shots offered no projection from Omicron’s variants [102], while another showed that a third booster shot provided more protection than two [103,104]. In a murine model, four vaccine shots were able to induce antibodies against the BA.1 and BA.2 variants. As more variants emerge, broadly neutralizing antibodies will be needed to quell all the variants of the SARS CoV-2 [105]. Overall, two doses of the vaccine were not effective at stopping the Omicron variants. In hospital care, according to Morais et al., the average cost for COVID-19 patients’ admission to hospital care was about $12,637.42. With Omicron being more transmissible, Omicron causes a lot of hospitalization, especially during January 2022, thereby impacting the hospital financially [89]. On 18 January 2023, Miller and colleagues reported preliminary data that the BQ.1.1 and XBB.1 subvariants of SARS-CoV-2 escaped neutralizing antibodies more effectively compared to the BA.5 variant by factors of 7 and 17, respectively, after monovalent mRNA vaccine boosting. Also, the BQ.1.1 and XBB.1 subvariants of SARS-CoV-2 escaped neutralizing antibodies more effectively than the BA.5 variant by factors of 7 and 21, respectively, after bivalent mRNA vaccine boosting [106].

## 7. Global Health Response and Future Variant Prediction

Globally, the dramatic and rapid spread of SARS-CoV-2 caused social changes that led to increased mental health burdens and fear-related behaviors. Physical and social distancing implemented throughout many parts of the world have significantly affected how the general population connects to and interacts with one another. Extreme avoidance of lack of social contact generated a sense of “loss of connection” that was further impaired by the inability to meet with friends and family at social gatherings and support areas, such as churches, restaurants, places of employment, and sports facilities. The closures of schools and many businesses, along with the rise in unemployment, further increased the sense of isolation, financial distress, anxiety, and depression [107]. In an article by Panchal et al., they found that children and adolescents with mental difficulties were more vulnerable to psychological distress due to the COVID-19 lockdown [108]. In countries such as Tanzania, the government-imposed lockdown has severely dismantled their trading business, which disrupted the country’s food supply [109]. In India, the lockdown was associated with an increased incidence of Tuberculosis [110]. Ironically, the government-imposed lockdown in Israel cost 50 to 466 times more money compared to their vaccination efforts [111]. On the other hand, the lockdown brought some benefits, including a significant decline in mortality due to car and/or travel-related accidents and a significant decrease in air pollution [112]. 

To minimize future pandemics, the U.S. Centers for Disease Control and Prevention’s global response to the COVID-19 pandemic has provided an overarching strategy to ensure health security for the American people. Strengthening the global healthcare systems to facilitate access to diagnostics and safe and effective therapeutics in response to COVID-19 can control the ongoing and minimize future COVID-19 pandemic threats [113]. 

However, such optimistic expectations can be ruined by the emergence of new variants, as in the case of the Omicron waves that emerged over the winter of 2020/2021. Even in countries that had managed to vaccinate a large fraction of their populations, the Omicron upsurge was inevitable. This, raises the question of whether it is possible to use scientific knowledge along with predictive mathematical models to anticipate changes and design management measures for these changes. In a recent article by Newcomb et al., they used data-driven modeling to show that the Omicron wave could have been predicted. Selection for these future variants tends to favor immune-escape variants due to the majority of the population becoming vaccinated or naturally immune from previous infections. However, this evolutionary process is slow and sporadic because individuals who have already been exposed to the virus or received vaccination have a broad immune response toward the infection [114]. Thereby, a single point mutation may not be sufficient for the virus to escape the immune response as pathogens do with modern drugs. Additionally, everyone has a unique immune system, and a virus that escapes one individual’s immune response may not do so in another individual immune response. Nonetheless, over time, new variants did emerge due to mutations within the S protein, and even worse, these variants do have a high mutation rate. In late 2021, the Omicron variant had about 34 mutations [115]. As with other coronavirus, reports have shown that antigenic drift can support the virus to render the antibody inadequate after a decade [114]. Further strengthening the argument that predictive models are warranted.

According to Dr. Newcomb, one can predict the future path of a pandemic depending on how the rates of vaccinations may interact with the new variants, the levels of social mitigation followed by a community, and the effectiveness and durations of the immunity generated by the current vaccines in a population [116]. Obermeyer, along with other scientists from the Broad Institute of MIT and Harvard, has developed a hierarchical Bayesian multinomial logistic regression model (PyR0). This mathematical model enables scalable analysis of the complete set of publicly available SARS-CoV-2 genomes, which can be applied to any viral genomic dataset and other viral phenotypes [117]. According to the author, PyR0 predicts the growth of new viral lineages based on their mutational profile; it ranks the fitness of lineages as new sequences when they become available and prioritizes mutations of biological and public health concerns for functional characterization. They used 6 million SARS-CoV-2 genomes obtained from the GISAID database in January 2022 and applied PyR0 to identify numerous substitutions that increase fitness, including previously identified spike mutations and many non-spike mutations within the nucleocapsid and nonstructural proteins. The PyR0 model was more accurate than naïve phylogenetic estimators. In another study by Maher et al., they developed a model with a high AUROC (0.92–0.97) score and found that ACE2 binding affinity was a useful predictor for mutational spread. Using their model, the global epidemiology metrics were more efficient at predicting VOC than local regions [118]. Such a tool can be extremely valuable, especially when dealing with the SARS-CoV-2 genome, which has accumulated many mutations. 

It is widely accepted that new viral lineages can increase the pathological potential of SARS-CoV-2, thereby making drug and vaccine development difficult [119]. Knowing the critical mutations can assist experts in predicting whether a new variant will be more transmissible or evade neutralizing antibodies by contributing to viral fitness. All of which can aid in the development of a better and an efficient target for future vaccines. Nevertheless, to address the public health emergency of COVID-19 and the emergence of new SARS-CoV-2 variants, alternative therapies are required. Recently, the use of a broad-spectrum antiviral compound was shown to be effective in controlling viral binding and replication [120]. In a study by Subhadra et al., they used the polyphenolic-rich compound Bi121, isolated from Pelargonium sidoides, and showed potential interference in the early steps of entry and replication of the B.1.167.2 (Delta) and Omicron SARS-CoV-2, using an in vitro cell culture model [120]. Although this approach seems promising, future studies are needed to provide its efficacy in patients with COVID-19.

## 8. Conclusions

After the vaccine rolled out in 2021, many economic pundits believed the economy would recover; however, Omicron’s variants emerged and cast doubt on the recovery. Among the different VOCs, the Omicron variant is the most genetically diverse and has evolved into several different sub-lineages that the WHO categorized into VOIs. BA.1, BA.1.1, BA.2, BA.2.12.1, BA.2.13, BA.2.38, BA.2.75, BA.3, BA.4, and BA.5 [121,122]. BA.4 and BA.5 are currently the most widespread and influential variants, while other novel subvariants, including BA.2.75.2(B.F 7), BA.4.6, BA.4.7, BA.5.9, BF.7, BQ.1, BQ.1.1, BN.1, XBB, XBB 1.5, XBB 1.6, and CH.1.1, evolved from various previously circulating sub-lineages of Omicron across the world [100]. The VOCs and VOIs have overwhelmingly impacted businesses, vaccine development, and hospital care. Many companies experienced high demand but also worker shortages due to employees falling ill. Concurrently, the U.S. government has spent over 19 billion dollars on private companies to develop the COVID-19 vaccine, but the Omicron VOIs emerged and were not neutralized by the vaccine. Experts predict that the XBB subvariant will not be the last new variant that emerges on the growing Omicron family tree. Moreover, medical experts are calling for a universal vaccine because they believe other types of coronaviruses will emerge since four fatal outbreaks have occurred [123]. As the world enters the fourth-year post-pandemic, the WHO said that COVID-19 remains a global health emergency. However, the world is transitioning into a new phase in which hospitalizations are reduced, and the number of deaths due to COVID has fallen to an ultimate low. On the other hand, as the Omicron sub-variants continue to predominate, their potency can decline and eventually become comparable to seasonal influenza, where the disease persists with milder symptoms. Moreover, the emergence of “layered immunity” as a result of reinfection should allow us to be better prepared for the next dominant strain. 

## Figures and Tables

**Figure 1 biology-12-01267-f001:**
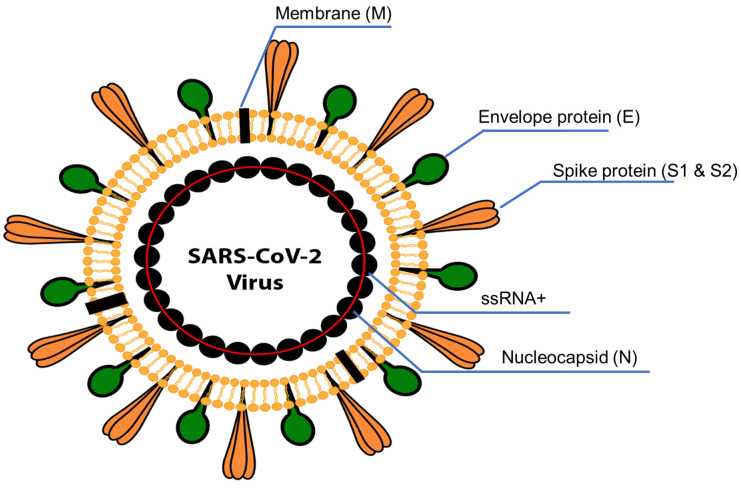
Schematic structure of SARS-CoV-2. The viral structure is primarily formed by the structural spike (S) proteins, envelope (E), membrane (M), and nucleocapsid (N) proteins. The S, M, and E proteins are all embedded in the viral envelope, a lipid bilayer derived from the host cell membrane. The N protein interacts with the viral RNA in the core of the virion.

**Figure 2 biology-12-01267-f002:**
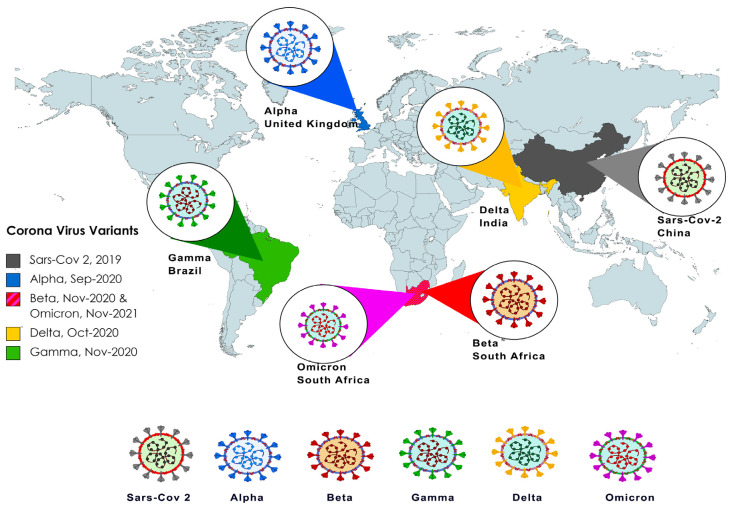
World map illustrating the years and the countries affected by the different VOC of SARS-CoV-2. Each VOC is color-coded for differentiation. Alpha (B.1.1.7) was the first VOC detected and it originated in the UK, followed by Beta (B.1.351), which originated in South Africa (SA) in August 2020. In April 2021, Delta (B.1.617.2) was detected in India. Omicron (B.1.1.529) is the most recent VOC identified and is divided into BA.1, BA.2, BA.4, and BA.5 lineages and several sub-lineages.

**Figure 3 biology-12-01267-f003:**
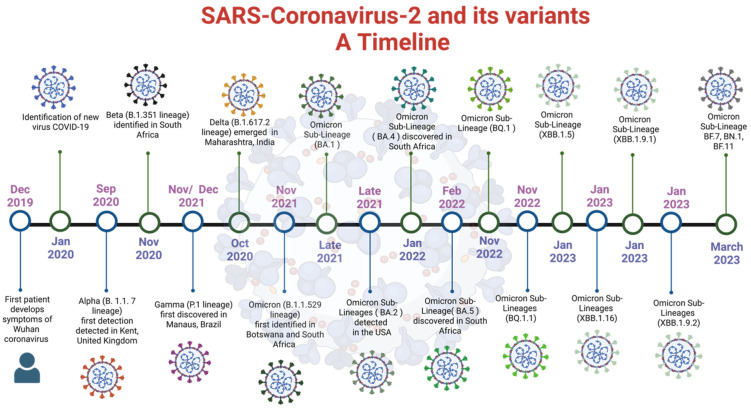
Timeline of different VOCs of SARS-CoV2. More than 17 different variants have been discovered starting from its origin in 2019 through the most recent VOC discovered early in 2023.

**Table 1 biology-12-01267-t001:** Major mutation of the spike protein in the VOCs and VOIs and their impact on viral features such as infectivity, transmissible nature, and response to treatment.

Variant	Key Mutations	Infectivity & Transmissibility	Response to Treatment	References Cited
**Alpha** (B.1.1.7)	N501Y, H69/V70, P681H, Y144	Increased transmissibility and virulence	Reduced efficacy of monoclonal antibodies and vaccines	[13,14,15]
**Beta** (B.1.351)	N501Y, E484K, K417N, L18F, D80A, D215G, A701V	Increased transmissibility	Reduced efficacy of monoclonal antibodies and vaccines	[16,17,18,19]
**Gamma** (P.1 lineage)	K417T, E484K, N501Y	Increased transmissibility and virulence	Reduced efficacy of monoclonal antibodies and vaccines, but previous infection provided between 54 to 79% protection	[20]
**Delta** (B.1.617)	R158G, L452R, T478K, D614G P681R, D950N	Increased infectivity and transmissibility compared to previous variants through the association of infected persons having a higher viral RNA load.	Treatment of monoclonal antibodies casirivimab, imdevimab and sotrovimab were associated with decreased hospitalizations and mortalities. Vaccination efficacy proved 80% effective at preventing symptomatic delta infection >240 after second vaccine dose.	[21,22,23]
**Omicron** (B.1.1.529)	Δ69–70 deletion, T95I, G142D/ Δ143–145 deletion, K417N, T478K, N501Y, N655Y, N679K, and P681H	Higher transmissibility with reduced case fatality compared to other variants.	Vaccination was ineffective at protection from symptomatic infection. Efficacious treatment modality was still largely restricted to glucocorticoids.	[24,25] [26,27] [28] [29]
**Subvariants** (BA.1, BA.1.1, BA.2, BA.4, BA.5, BQ.1, BQ.1.1, XBB.1.5) BA2.75 BA.4 and BA.5 XBB.1.5 XBB.1.16 BQ.1 and BQ.1.1	>60 mutations >50% mutations reside in the *spike* gene (~30 substitutions, 6 residue deletions, 3 residue insertions); other mutations dominate the RBD and NTD. W152R, F157L, I210V, G257S, D339H, G446S, N460K, Q493 Δ69–70 deletion, L452R, F486V, and R493Q N460K, S486P, F490S E180V, T478R, F486P K444T, L452R, N460K, and F486V Y273H (NSP12), N268S (NSP13)	High infectivity and transmissibility; most dominant strain. BA.4 and BA.5 more infectious than BA.2. Antibody evasion and enhanced viral fitness.	Insufficient neutralization of Omicron antibodies from current COVID-19 vaccines. Enhanced neutralization resistance Resistant to the monoclonal antibody drug therapy	

## Data Availability

Not applicable.

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
