# Peer review of "From Alpha to Omicron: How Different Variants of Concern of the SARS-Coronavirus-2 Impacted the World"

_biology, 2023, doi:10.3390/biology12091267_

Round 1
Reviewer 1 Report
Andre and colleagues comprehensively reviewed various SARS-CoV-2 VOCs and how they impacted the world.
Line 133: Figure 2 illustrates the years and countries affected by different VOCs of SARS-CoV-2. Timeline of VOCs (e.g. Beta, Omicron) in Figure 2 do not match the timeline presented in Figure 3.
Line 252: D614G "mutation showed lower Ct values in PCR testing, suggesting an underestimation of viral load in patients". Further explanation required as Korber et. al. (2020) reported G614 is associated with lower PCR cycle thresholds (suggesting higher viral loads)?
It would be appropriate to add a summary table to compare different VOCs (in reference to the ancestral strain), in terms of key mutations, infectivity and transmissibility, virulence, response to treatment/vaccination, antibody evasion/viral neutralization, morbidity, etc.
Author Response
We would like to thank the reviewers for their insightful comments and have responded to each in blue font in the body of the manuscript. We were pleased with the overall outcome and enthusiasm among the reviewer panel and feel that the comments and suggested provided by the reviewers strengthened the manuscript.
REVIEWER #1:
Comment 1: Line 133: Figure 2 illustrates the years and countries affected by different VOCs of SARS-CoV-2. Timeline of VOCs (e.g. Beta, Omicron) in Figure 2 do not match the timeline presented in Figure 3.
Response: We thank the reviewer for his/her careful evaluation, and we apologize for the oversight. The dates have been corrected.
Comment 2: Line 252: D614G "mutation showed lower Ct values in PCR testing, suggesting an underestimation of viral load in patients". Further explanation required as Korber et. al. (2020) reported G614 is associated with lower PCR cycle thresholds (suggesting higher viral loads)? Response: The following paragraph has been added: “Moreover, the global replacement of the G614 variant with the D614 variant in SARS-CoV-2-infected individuals showed a shift to a more dominant form of the pseudovirus. According to Korber et al., spike G614 in human patients, was associated with greater infectivity and higher levels of viral nucleic acid in the upper respiratory tract, despite lower RT-PCR cycle thresholds. Suggesting higher viral loads, but not with increased disease severity”.
Comment 3: It would be appropriate to add a summary table to compare different VOCs (in reference to the ancestral strain), in terms of key mutations (mutations of S protein of different VOCs), infectivity and transmissibility, virulence, response to treatment/vaccination, antibody evasion/viral neutralization, morbidity, etc.
Response: A summary Table 1, has been included in the text, and provides the key mutations, infectivity and transmissibility, response to treatment/vaccination between different VOCs and the subvariants of the Omicron strain.
Reviewer 2 Report
Overall Review:
1. The paper presents a comprehensive discussion on the evolution of the SARS-CoV-2 virus since its initial outbreak in 2019.
2. It provides insights into the genetic mutations and the emergence of different Variants of Concern (VOC) and their classifications.
3. The authors emphasize the properties of these variants, highlighting their increased transmissibility, virulence, and potential for immune evasion.
Strength:
1. In-depth Analysis: The paper offers a thorough examination of the SARS-CoV-2 virus's genetic evolution, providing readers with a clear understanding of its progression over time.
2. Relevance to Current Events: Given the global impact of COVID-19, understanding the various variants and their implications is crucial. This paper serves as a timely resource in this regard.
3. Clear Classification: The paper presents a clear categorization of the variants as per the WHO classifications, aiding in the comprehension of their significance and implications.
Concerns:
1. Could you provide more insights into the real-world impact of these variants, especially in terms of global health responses and challenges faced?
2. How do these variants affect the efficacy of current vaccines, and what implications does this have for future vaccine development?
3. Are there any predictive models or insights into potential future variants and their characteristics?
4. Some important references should be cited
Author Response
We would like to thank the reviewers for their insightful comments and have responded to each in blue font in the body of the manuscript. We were pleased with the overall outcome and enthusiasm among the reviewer panel and feel that the comments and suggested provided by the reviewers strengthened the manuscript.
REVIEWER #2:
Comment #1: Could you provide more insights into the real-world impact of these variants, especially in terms of global health responses and challenges faced?
Response: A new section on Global Health Response & Future Variant Predication has been included and reads as follows:” Globally, the dramatic and rapid spread of SARS-CoV-2 caused social changes that lead to increased mental health burden and fear-related behaviors. Physical and social distancing implemented throughout many parts of the world have significantly affected how the general population connects to and interacts with one another. Extreme avoidance of lack of social contact generated a sense of “loss of connection” that was further impaired by the inability to meet with friends and family at social gathering and support areas such as churches, restaurants, place of employment and sports facilities. The closures of schools and of many businesses along with the rise of unemployment further increased the sense of isolation, financial distress anxiety and depression [103]. In an article by Panchal et al., they found that children and adolescents with mental difficulties were more vulnerable to psychological distress due to the COVID-19 lockdown [104]. In countries such as Tanzania, the government-imposed lockdown has severely dismantled their trading business, which disrupted the country food supply [105]. In India, the lockdown was associated with increased incidence of Tuberculosis [106]. Ironically, the government-imposed lockdown in Israel cost 50 to 466 times more money compared to their vaccination efforts [107]. On the other hand, the lockdown brought some benefits, including a significant decline in mortality due to car and / or travel-related accidents and a significant decrease in air pollution [108].
To minimize future pandemic, the U.S. Centers for Disease Control and Prevention’s global response to the COVID-19 pandemic has provided an overarching strategy to ensure health security to the American people. Strengthening the global health care systems to facilitate access to diagnostics, safe and effective therapeutics in response to COVID-19, can control the ongoing and minimize future COVID-19 pandemic threats (CDC.gov). However, such optimistic expectation can be ruined by the emergence of new variants, as in the case of the omicron waves that emerged over the winter of 2020/2021. Even in countries that had managed to vaccinate a large fraction of their populations, the omicron upsurge was inevitable. Thus, raising the question about the whether it is possible to use scientific knowledge along with predictive mathematical models to anticipate changes and design management measures for these changes. In a recent article by Newcomb et al., they used data-driven modelling to show that the omicron wave could have been predicted. Selection for these future variants tend to favor immune-escape variants due to majority of the population becoming vaccinated or naturally immune from previous infection. However, this evolutionary process is slow and sporadic because individuals who have already been exposed to the virus or received vaccination have a broad immune response toward the infection [109]. Thereby, a single point mutation may not be sufficient for the virus to escape the immune response as pathogens do with modern drugs. Additionally, everyone has a unique immune system, and a virus that escapes one individual’s immune response, may not do so in another individual immune response. Nonetheless, overtime new variants did emerge due to mutations within the S protein, and even worse, these variants do have a high mutation rate. In late, 2021 the Omicron variant had about 34 mutations [110]. As with other coronavirus, report have showed that antigenic drift can support the virus to render the antibody inadequate after a decade [109]. Further strengthening the argument that predictive models are warranted.
According to Dr. Newcomb, one can predict the future path of a pandemic de-pending on how the rates of vaccinations may interact with the new variants, the levels of social mitigation followed by a community, and the effectiveness and durations of the immunity generated by the current vaccines in a population [111]. Obermeyer along with other scientists from the Broad Institute of MIT and Harvard have developed a hierarchical Bayesian multinomial logistic regression model (PyR0). This mathematical model enables scalable analysis of the complete set of publicly available SARS-CoV-2 genomes, that can be applied to any viral genomic dataset and to other viral phenotypes [112]. According to the author, PyR0 predicts the growth of new viral lineages based on their mutational profile, it ranks the fitness of lineages as new sequences when they become available and prioritizes mutations of biological and public health concern for functional characterization. They used 6 million SARS-CoV-2 genomes obtained from the GISAID database in January 2022, and applied PyR0 to identify numerous substitutions that increase fitness, including previously identified spike mutations and many non-spike mutations within the nucleocapsid and nonstructural proteins. The PyR0 model was more accurate than naïve phylogenetic estimators. In another study, by Maher et al., they developed a model with a high AUROC (0.92-0.97) score and found that ACE2 binding affinity was a useful predictor for mutational spread. Using their model, the global epidemiology metrics were more efficient at predicting VOC than local regions [113]. Such a tool can be extremely valuable, especially when dealing with the SARS-CoV-2 genome, that now has accu-mulated many mutations.
It is widely accepted that new viral lineages can increase the pathological potential of SARS-CoV-2, thereby making drug and vaccine development difficult [114]. Knowing the critical mutations can assist experts in predicting whether a new variant will be more transmissible or evade neutralizing antibodies by contributing to the viral fitness. All of which can aid in the development of a better and an efficient target for future vaccines. Nevertheless, to address the public health emergency of COVID-19 and emerging of new SARS-CoV-2 variants, alternative therapies are required. Recently, the use of broad-spectrum antiviral compound was shown effective in controlling viral binding and replication [115]. In a study by Subhadra et al., they used the polyphenolic-rich compound, Bi121, isolated from Pelargonium sidoides, and showed potential interference in the early steps of entry and replication of the B.1.167.2 (Delta) and Omicron SARS-CoV-2, using in vitro cell culture model [115]. Although this approach seems promising, future studies are needed to provide its efficacy in patients with COVID.”
Comment #2. How do these variants affect the efficacy of current vaccines, and what implications does this have for future vaccine development?
Response: The efficacy of current vaccines, and implications for future vaccine development has been summarized in Table 1
Comment #3. Are there any predictive models or insights into potential future variants and their characteristics?
Response: A paragraph related to predictive models or insights into potential future variants has been included under section 7. Global Health Response & Future Variant Predication and reads as follows:” In a recent article by Newcomb et al., they used data-driven modelling to show that the omicron wave could have been predicted. Selection for these future variants tend to favor immune-escape variants due to majority of the population becoming vaccinated or naturally immune from previous infection. However, this evolutionary process is slow and sporadic because individuals who have already been exposed to the virus or received vaccination have a broad immune response toward the infection [109]. Thereby, a single point mutation may not be sufficient for the virus to escape the immune response as pathogens do with modern drugs. Additionally, everyone has a unique immune system, and a virus that escapes one individual’s immune response, may not do so in another individual immune response. Nonetheless, overtime new variants did emerge due to mutations within the S protein, and even worse, these variants do have a high mutation rate. In late, 2021 the Omicron variant had about 34 mutations [110]. As with other coronavirus, report have showed that antigenic drift can support the virus to render the antibody inadequate after a decade [109]. Further strengthening the argument that predictive models are warranted.
According to Dr. Newcomb, one can predict the future path of a pandemic de-pending on how the rates of vaccinations may interact with the new variants, the levels of social mitigation followed by a community, and the effectiveness and durations of the immunity generated by the current vaccines in a population [111]. Obermeyer along with other scientists from the Broad Institute of MIT and Harvard have developed a hierarchical Bayesian multinomial logistic regression model (PyR0). This mathematical model enables scalable analysis of the complete set of publicly available SARS-CoV-2 genomes, that can be applied to any viral genomic dataset and to other viral phenotypes [112]. According to the author, PyR0 predicts the growth of new viral lineages based on their mutational profile, it ranks the fitness of lineages as new sequences when they become available and prioritizes mutations of biological and public health concern for functional characterization. They used 6 million SARS-CoV-2 genomes obtained from the GISAID database in January 2022, and applied PyR0 to identify numerous substitutions that increase fitness, including previously identified spike mutations and many non-spike mutations within the nucleocapsid and nonstructural proteins. The PyR0 model was more accurate than naïve phylogenetic estimators. In another study, by Maher et al., they developed a model with a high AUROC (0.92-0.97) score and found that ACE2 binding affinity was a useful predictor for mutational spread. Using their model, the global epidemiology metrics were more efficient at predicting VOC than local regions [113]. Such a tool can be extremely valuable, especially when dealing with the SARS-CoV-2 genome, that now has accumulated many mutations.
Comment #4. Some important references should be cited.
Response: Per request of the reviewer, additional (what we believe are important) references have been included in the references list in the manuscript and are listed below.
- Walensky, R.P., H.T. Walke, and A.S. Fauci, SARS-CoV-2 Variants of Concern in the United States-Challenges and Opportunities. Jama, 2021. 325(11): p. 1037-1038.
- Arbel, R. and J. Pliskin, Vaccinations versus Lockdowns to Prevent COVID-19 Mortality. Vaccines (Basel), 2022. 10(8).
- Borri, N., et al., The "Great Lockdown": Inactive workers and mortality by Covid-19. Health Econ, 2021. 30(10): p. 2367-2382.
- CDC. CDC Strategy for Global Response to COVID-19 (2020-2023). Nov. 10, 2022 [cited 2023 Sept 09]; Available from: https://www.cdc.gov/coronavirus/2019-ncov/global-covid-19/global-response-strategy.html#:~:text=public%20health%20emergencies.-,Goals,global%20public%20health%20leadership%3B%20and.
- Otto, S.P., et al., The origins and potential future of SARS-CoV-2 variants of concern in the evolving COVID-19 pandemic.Curr Biol, 2021. 31(14): p. R918-r929.
- Magazine, N., et al., Mutations and Evolution of the SARS-CoV-2 Spike Protein. Viruses, 2022. 14(3).
- Newcomb, K., S. Bilal, and E. Michael, Combining predictive models with future change scenarios can produce credible forecasts of COVID-19 futures. PLoS One, 2022. 17(11): p. e0277521.
- Obermeyer, F., et al., Analysis of 6.4 million SARS-CoV-2 genomes identifies mutations associated with fitness. Science, 2022. 376(6599): p. 1327-1332.
- Maher, M.C., et al., Predicting the mutational drivers of future SARS-CoV-2 variants of concern. Sci Transl Med, 2022. 14(633): p. eabk3445.
- Morens, D.M., J.K. Taubenberger, and A.S. Fauci, Universal Coronavirus Vaccines - An Urgent Need. N Engl J Med, 2022. 386(4): p. 297-299.
Reviewer 3 Report
In this manuscript, Andre and colleagues summarized the genetic origin and genetic makeup, epidemiology, response to treatment/vaccination of SARS-CoV-2 VOC variants from Alpha to Omicron. This review provides us with a comprehensive evolution picture of SARS-CoV-2. After reviewing, I have some suggestions for authors.
Major revision Suggestions:
1. In March 2023, WHO updated its tracking system and working definitions for variants of concern, variants of interest and variants under monitoring.
(https://www.who.int/news/item/16-03-2023-statement-on-the-update-of-who-s-working-definitions-and-tracking-system-for-sars-cov-2-variants-of-concern-and-variants-of-interest)
“The Omicron viruses account for over 98% of the publicly available sequences since February 2022 and constitute the genetic background from which new SARS-CoV-2 variants will likely emerge, although the emergence of variants derived from previously circulating VOCs or of completely new variants remains possible. The previous system classified all Omicron sublineages as part of the Omicron VOC and thus did not have the granularity needed to compare new descendent lineages with altered phenotypes to the Omicron parent lineages (BA.1, BA.2, BA.4/BA.5). Therefore, from 15 March 2023, the WHO variant tracking system will consider the classification of Omicron sublineages independently as variants under monitoring (VUMs), VOIs, or VOCs. WHO will assign Greek labels for VOCs, and will no longer do so for VOIs. With these changes factored in, Alpha, Beta, Gamma, Delta as well as the Omicron parent lineage (B.1.1.529) are considered previously circulating VOCs. WHO has now classified XBB.1.5 as a VOI.”
Therefore, the author should update the relevant information in the manuscript.
2. Figure 2 is not clear and intuitive at present, it is suggested to re-create the temporal and spatial distribution of VOCs in Figure 2. For example, it can be filled with different colors to indicate the geographical distribution of different variants.
3. It is suggested to add schematics to show the mutations of S protein of different VOCs, especially those in the RBD region.
4. It is suggested to add a section of progress about broad-spectrum treatments for different VOC variants, including broad-spectrum vaccines, broad-spectrum antibody drugs and other research progress.
Minor revision Suggestions:
1. Line 30-31: check the grammar.
2. Line 42: “receptor” may be “as receptor”.
3. Line 47, Line 51 and Line 252: “ACE2 receptor” may be “ACE2”.
4. Line 404: “from Hunan”, check the word. “Wuhan” or “Human”?
5. Line 426-427: “reduced to 78.7%” should be “78.7% reduced” or “reduced by 78.7%.
Author Response
We would like to thank the reviewers for their insightful comments and have responded to each in blue font in the body of the manuscript. We were pleased with the overall outcome and enthusiasm among the reviewer panel and feel that the comments and suggested provided by the reviewers strengthened the manuscript.
REVIEWER # 3:
Comment #1. In March 2023, WHO updated its tracking system and working definitions for variants of concern, variants of interest and variants under monitoring. Therefore, the author should update the relevant information in the manuscript.
Response: We have updated the information as follows: Recently, the WHO has designated XBB.1.5 and XBB.1.16. as variants of interest (VOI).
Comment #2. Figure 2 is not clear and intuitive at present, it is suggested to re-create the temporal and spatial distribution of VOCs in Figure 2. For example, it can be filled with different colors to indicate the geographical distribution of different variants.
Response: Different colors have been added to indicate the geographical areas.
Comment #3. It is suggested to add schematics to show the mutations of S protein of different VOCs, especially those in the RBD region.
Response: Different mutations off the S protein, especially those in the RBD region have been included summary Table 1.
Comment #4. It is suggested to add a section of progress about broad-spectrum treatments for different VOC variants, including broad-spectrum vaccines, broad-spectrum antibody drugs and other research progress.
Response: A section of progress about broad-spectrum treatments have been included under section 7. Global Health Response & Future Variant Predication and it reads as follows:” It is widely accepted that new viral lineages can increase the pathological potential of SARS-CoV-2, thereby making drug and vaccine development difficult [114]. Knowing the critical mutations can assist experts in predicting whether a new variant will be more transmissible or evade neutralizing antibodies by contributing to the viral fitness. All of which can aid in the development of a better and an efficient target for future vaccines. Nevertheless, to address the public health emergency of COVID-19 and emerging of new SARS-CoV-2 variants, alternative therapies are required. Recently, the use of broad-spectrum antiviral compound was shown effective in controlling viral binding and replication [115]. In a study by Subhadra et al., they used the polyphenolic-rich compound, Bi121, isolated from Pelargonium sidoides, and showed potential interference in the early steps of entry and replication of the B.1.167.2 (Delta) and Omicron SARS-CoV-2, using in vitro cell culture model [115]. Although this approach seems promising, future studies are needed to provide its efficacy in patients with COVID”.
Comment #5: Minor changes
- Line 30-31: check the grammar. Response: Sentence has been checked
- Line 42: “receptor” may be “as receptor”. Changed to “as receptor.”
- Line 47, Line 51 and Line 252: “ACE2 receptor” may be “ACE2”. Changed to “ACE2”.
- Line 404: “from Hunan”, check the word. “Wuhan” or “Human”? Changed to “Wuhan.”
- Line 426-427: “reduced to 78.7%” should be “78.7% reduced” or “reduced by 78.7%. Changed to “reduced by 78.7%”